# THE FRÉCHET DISTANCE OF TRAINING AND TEST DISTRIBUTION PREDICTS THE GENERALIZATION GAP

## ABSTRACT

Learning theory tells us that more data is better when minimizing the generalization error of identically distributed training and test sets. However, when training and test distribution differ, this distribution shift can have a significant effect. With a novel perspective on function transfer learning, we are able to lower bound the change of performance when transferring from training to test set with the Wasserstein distance between the embedded training and test set distribution. We find that there is a trade-off affecting performance between how invariant a function is to changes in training and test distribution and how large this shift in distribution is. Empirically across several data domains, we substantiate this viewpoint by showing that test performance correlates strongly with the distance in data distributions between training and test set. Complementary to the popular belief that more data is always better, our results highlight the utility of also choosing a training data distribution that is close to the test data distribution when the learned function is not invariant to such changes.

## 1 INTRODUCTION

Imagine there are two students who are studying for an exam. Student A studies by diligently learning the class material by heart. Student B studies by learning the underlying reasons for why things are the way they are. Come test day, student A is only able to answer test questions that are very similar to the class material while student B has no trouble answering different looking questions that follow the same reasoning. Distilled from this example, we note there is a trade-off between how "well" a student studied, i.e., how indifferent the student is to receiving exercise or test questions, and how close the test questions are to the exercise questions.

While most machine learning work studies the generalization error, i.e., the error when testing on different samples from the *same* distribution, we do not take the match of train and test distribution as given. In fact, it appears that the distance between train and test distribution may be critical for successful "generalization". Following a similar line of thought, Uguroglu & Carbonell (2011) devised a distribution measurement to select only features that do not vary from one domain to another. In contrast, we are interested in linking performance directly to the distance between train and test distribution.

*Invariance to distribution shifts:* We say that a function is *invariant* to a given input perturbation when the corresponding output does not change with the perturbation. This is desirable when trying to achieve robustness to irrelevant data variations which are called *nuisances* (Achille & Soatto, 2018). As outlined by Achille & Soatto (2018); Censi & Murray (2011), the "optimal" learned function from input to output is maximally invariant to all data variations that do not contain information about the output. To the extent to which a learner reacts to such nuisance variations, which carry no information about the output, it will incur a performance change in expectation. The difficulty lies in knowing what can be ignored and what cannot.

*Similarity between training and test distribution:* Another strategy would be to ensure that the training and test distribution match which has been investigated in a number of diverse settings (Tzeng et al., 2017; Arjovsky et al., 2017). Variations of this theme were encountered by Zhang et al. (2016), where they show that networks are able to fit random labels perfectly, yet understandably fail to generalize to the test set of the correct label distribution.

*Contribution:* We frame the learning problem of training on a training set then shifting to test on a test set as a transfer learning problem that goes beyond the usual generalization which assumes the same distribution for training and testing. Based on our analysis, we aim to *explicitly* express the trade-off of the distance between both data set distributions given by the Wasserstein distance, which measures how close training and testing distribution are, and how *invariant* the learned function is to the training to test distribution shift. Joined together this is expressed via the Wasserstein distance between training and test samples embedded in the feature space of the learned function. Our experiments show a strong negative linear correlation between the distribution distances and network performance. This corroborates the notion that as long as the true function class is not found, it is best to ensure that training and test distribution are close. While this conclusion seems intuitive, it may often be overlooked in the hunt for more data.

## 2 RELATED WORK

The trade-off and effect on performance between invariance to distribution shifts and the magnitude of such distribution shifts has been an active field of research, albeit mostly implicitly. Given the correct invariance, changes in distribution do not have a strong effect and learning "generalizes". What has not been investigated so far, to the best of our knowledge, is a more systematic treatment of measuring distribution distances and invariances of learned functions. These assumptions currently remain mainly implicit in the literature.

**Invariance:** Invariance is a key component of Deep Learning (Goodfellow et al., 2009; Achille & Soatto, 2018) with many works focusing on increasing invariance by incorporating structure into their models (LeCun et al., 1998; Bruna & Mallat, 2012; Cohen et al., 2018; Kumar et al., 2017; van der Wilk et al., 2018). Other works apply GANs (Goodfellow et al., 2014), e.g., to domain adaptation problems (Tzeng et al., 2017; Hoffman et al., 2017), to map source and target distributions to the same space while being invariant to occurring distribution shifts. As stand in for many similar works, Tzeng et al. (2015) employ invariance to improve visual adaptation to domain changes. In this work, rather than trying to beat a benchmark, we aim to establish the quantitative and qualitative trade-offs between invariance and distribution shift. In contrast to adapting the structure of a model, many works achieve invariance through data augmentation (Krizhevsky et al., 2017b; Shorten & Khoshgoftaar, 2019; Hernández-García et al., 2019). While most augmentations are handpicked based on domain knowledge, automated methods based on GANs have been successfully applied to data augmentation (Antoniou et al., 2017; Kim et al., 2017). Any of these transformations can be considered a *canonization* (Censi & Murray, 2011), i.e., a generalization of data normalization to group actions, by removing unwanted nuisances to increase robustness. Many other techniques such as regularization Leen (1995), and dimension reduction can also be considered part of the umbrella of invariance learning (Jolliffe & Springer-Verlag, 2002; Hadsell et al., 2006; Saul et al., 2006). In this work, we simply take the inherent invariance of learned embeddings to approximate the invariance of our function approximator and focus on the distributional shift from train to test distribution.

**Distribution shifts:** Shifts in data set distribution (Storkey, 2009; Quionero-Candela et al., 2009) can take a number of forms where either the input distribution $p(x)$, the target distribution $p(y)$ or a combination thereof changes. Likewise imbalanced datasets (Japkowicz & Stephen, 2002), methods of over and undersampling (Bowyer et al., 2011), and domain shifts (Zhang et al., 2013) can be described in this framework. By knowing how the distribution changes from one setting to another, Zanardi et al. (2019) were able to define an operating domain for different vision sensors and employ this for a semi-supervised learning setup. Part of evaluating distribution shifts is measuring these probability distribution distances. A popular use for this is the Maximum Mean Discrepancy (MMD) distance (Borgwardt et al., 2006) that has found application in anomaly detection (Zou et al., 2017) and two-sample testing (Gretton et al., 2012). Other distances such as the Wasserstein distance have made an impact in computational graphics modeling (Solomon et al., 2015) and in generating realistic-looking synthetic images using GANs (Arjovsky et al., 2017). In this work, we apply the Wasserstein distance as a powerful tool to investigate the performance drop when moving from train to test samples.

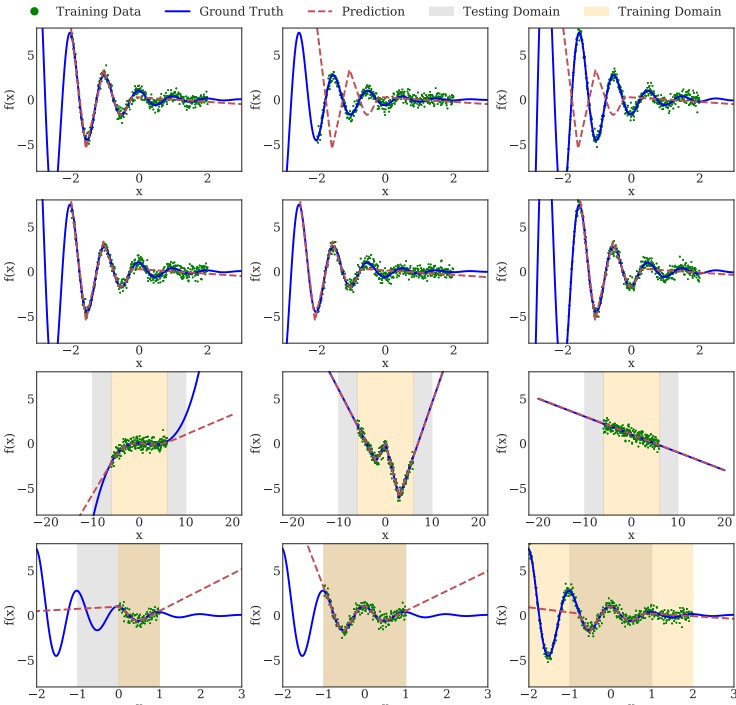

Figure 1: Function approximations of a 2 layer neural network with changing training and testing domains. (First row) Testing domain input shifted by the scalar $\alpha = [0, -0.5, 0.5]$ from left to right respectively. (Second row) Similar configuration as the first row, but with an additional zero mean normalization on the input data. (Third row) Function approximations of a nonlinear, piecewise linear and linear function (left to right). (Fourth row) The training set domain is either smaller, equal, or larger than the test domain. Best viewed in color.

## 3 INSIGHTS FROM SMALL-SCALE EXPERIMENTS

Before jumping ahead, we gain an intuitive understanding of the behavior of a neural network in the case of mismatched train and test distributions. In particular we are interested in the effect of distribution shift from training to test distribution and the effect of invariance of the learned function to such changes. Looking at this through a different lens, we also investigate what impact the type of function we are approximating, and the type of function we are approximating with can have.

First, we run experiments detailing changes in the input distribution as illustrated in the top row of Fig. 1. From left to right, we increase input values x by adding a scalar $\alpha$. As can be readily observed, the quality of the function approximation deteriorates under such a distribution shift. If on the other hand, we normalize the input data by subtracting the mean of the distribution, the function approximation remains unchanged as seen in the second row of Fig. 1. This added invariance through normalization made the function approximation robust to the given distribution shift.

Secondly, we run function approximation experiments in which we have a training set in one interval and a test set in another. We focus on what happens when the true function class is the same as the one used for approximation and what happens when this is not the case. We observe in the third row of Fig. 1 that when the correct function is found, as in the second and third image, the approximated function works even in a domain in which it was not trained. On the other hand, the first plot of the third row and the plots in the fourth row show that if the true function is not found exactly, then outside of the training domain errors accumulate. Further, we conclude from the bottom row in Fig. 1 that a oversized training domain (bottom right plot) can hurt performance on the test domain, when compared to the middle plot with overlapping train and test domains.

Our takeaway is that invariance indeed influences how distribution shifts affect changes in training and testing performance. We however also note that this topic is more nuanced. The difficulty lies in the fact that finding the "correct" invariances also means finding the true function as visualized in the third row of Fig. 1. These smaller experiments make us expect that it may be unlikely to find the true invariances of an underlying functional relationship. In such a case, ensuring that train and test distribution match thus becomes a viable option to enforce a small change from train to test performance.

# 4   THE TRADE-OFF BETWEEN INVARIANCE AND DISTRIBUTION SHIFT

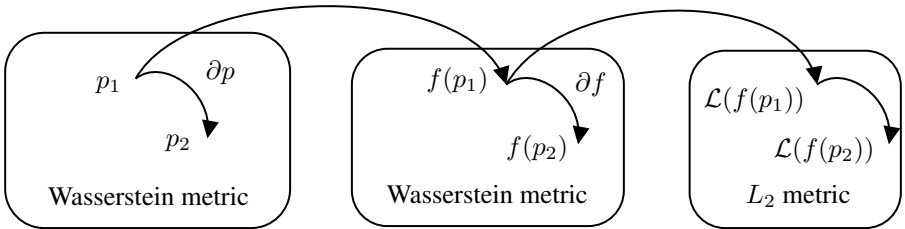

Figure 2: Illustration of the relationship between dataset distributions $p_i$ to learned induced distribution $f(p_i)$ to associated expected loss $\mathcal{L}(f(p_i))$. Noted at the bottom of the spaces are the assumed underlying metrics of the space.

We begin our investigation by stating that learning is a process that maps a training dataset $\mathcal{D}_{\text{train}}$ with distribution $p_{\text{train}}(x, y)$ to a learned function approximation $f(p_{\text{train}}, x) \approx y \ \forall (x, y) \in D_{\text{train}}$, inducing a target distribution $f(p_{\text{train}})$ as depicted in Fig. 2. In deep learning this is usually achieved by minimizing a loss function on the training set using stochastic gradient descent (Robbins & Monro, 1951). Of interest however is the performance of the learned function on a testing dataset $\mathcal{D}_{\text{test}}$ with possibly different distribution $p_{\text{test}}(x, y)$. By modeling the learned relationship $f(p_{\text{train}}, x) \in Y$ as a function on the data distribution $p_{\text{train}}(x, y)$, we are then able to obtain a relationship between changes in the data distribution and changes in the learned function and its performance.

**Theorem 1.** *We are given two dataset distributions $p_1(x, y)$ and $p_2(x, y)$ of input data $x \in X$ and output $y \in Y$ situated in the probability space with Wasserstein metric $W_2(\cdot, \cdot)$ and assume that $f(p, x) : P \times X \to Y$ is the function of a function family $\mathcal{F}$ continuous in the probability space $P$ that minimizes the norm loss $\mathcal{L}(f(p_1, x), y) = ||f(p_1, x) - y||_2$ when the data is distributed according to $p_1(x, y)$. Then the expected absolute difference in loss for distribution $p_2$ has the following lower bound.*

$$\mathbb{E}_{p_2} |\mathcal{L}(f(p_2, x), y) - \mathcal{L}(f(p_1, x), y)| \geq \mathbb{E}_{p_2} \left|\left| \overline{\frac{\partial f(p, x)}{\partial p}} \right|\right| \cdot W_2(p_1, p_2) \tag{1}$$

*where $\overline{\frac{\partial f(p,x)}{\partial p}}$ denotes the average value of $\frac{\partial f(p,x)}{\partial p}$ along the integral as part of the intermediate value theorem.*

*Proof.* First we note that we can write the learned function for distribution $p_2$ as the function learned for distribution $p_1$ plus the line integral of the change of the function along the Wasserstein geodesic between $p_1$ and $p_2$.

(i)   $$f(p_2, x) = f(p_1, x) + \int_{p_1}^{p_2} \frac{\partial f(p, x)}{\partial p} \partial p, \quad \partial p := \nabla_p W_2(p, p_1)$$

(ii)   $$\int_{p_1}^{p_2} \partial p = \int_{p_1}^{p_2} \nabla_p (W_2(p, p_1)) dp = W_2(p_2, p_1) - W_2(p_1, p_1) = W_2(p_2, p_1)$$

Then the change in loss is lower bounded as follows:

$$
\begin{aligned}
\mathbb{E}_{p_2}\Delta\mathcal{L} &= \mathbb{E}_{p_2}|\mathcal{L}(f(p_2,x),y) - \mathcal{L}(f(p_1,x),y)| \\
&= \mathbb{E}_{p_2}\Big|\,||f(p_2,x) - y|| - ||f(p_1,x) - y||\,\Big| \geq \mathbb{E}_{p_2}||f(p_2,x) - f(p_1,x)|| \\
&\stackrel{(i)}{=} \mathbb{E}_{p_2}\Big|\Big|\int_{p_1}^{p_2}\frac{\partial f(p,x)}{\partial p}\partial p\Big|\Big| \stackrel{\text{Intermediate value}}{=} \mathbb{E}_{p_2}\Big|\Big|\overline{\frac{\partial f(p,x)}{\partial p}}\int_{p_1}^{p_2}\partial p(x)\Big|\Big| \\
&\stackrel{(ii)}{=} \mathbb{E}_{p_2}\Big|\Big|\overline{\frac{\partial f(p,x)}{\partial p}}W_2(p_1,p_2)\Big|\Big| = \mathbb{E}_{p_2}\Big|\Big|\overline{\frac{\partial f(p,x)}{\partial p}}\Big|\Big|\cdot W_2(p_1,p_2)
\end{aligned}
$$

$\square$

From the above lower bound in Ineq. 1, we can deduce two ways of achieving a minimal change in loss when moving from training distribution $p_1$ to testing distribution $p_2$.

**(1)** Either the difference in distribution from training set to test set is small $W_2(p_1,p_2) \approx 0$. In this case, by assuming the learned function fits the training distribution perfectly and is not ill-conditioned, we can deduce that also the test distribution is fit perfectly, or

**(2)** The function $f(p)$ does not change when the dataset distribution changes from $p_1$ to $p_2$ by being invariant to such changes s.t. $\mathbb{E}_{p_2}\big|\big|\frac{\partial f(p,x)}{\partial p}\big|\big| \approx 0$.

Clearly, if the learned function cannot distinguish between, i.e. is invariant to, changes in training and test set, then even a shift of distribution to the test set cannot alter the quality of the performance. If on the other hand, as may be more common, the function class is not invariant, then the difference in distribution from training to test set becomes decisive. Note that since we regard changes in performance, it is vital to already have a good performance on the training distribution. From the above analysis, we believe that it may be essential to employ training sets that are as close as possible in distribution to the test set. If this is the case one does not need to worry about finding the right function class with the right invariances to approximate the true function. In the case where the learned function has "too much invariance", the function is not noticing some of the relevant variations in the data, e.g. in the extreme a constant would not change from training to testing distribution. While its performance does not change from training to test distribution, its training performance already suffers from unnecessary invariance. If on the other hand, the function is not invariant enough it might fit the training set well yet lose performance when switching to the test distribution.

Instead of either calculating the distribution distance on the raw data ($W_2(p_1,p_2)$), we compute the distance on an embedding of the raw data that aims to mirror the distances the function $f(\cdot)$ would perceive. This has the advantage of avoiding to compute demanding high dimensional distribution distances from raw data on the one hand, and having to train networks for both training and test distribution on the other hand. Additionally, it may be hard to properly compare the distribution of a function trained on the training distribution vs. one trained on the test distribution since training and test set oftentimes are of very different size.

To measure distribution distances, we employ the Fréchet distance (FD) (Fréchet, 1957), also called the 2-Wasserstein distance (Vaserstein, 1969), by assuming all probabilities to be normally distributed. The FD is a distance measure between two normal distributions, also adopted in the *Fréchet Inception Distance* (FID) (Heusel et al., 2017) to evaluate the quality difference of real and synthetic images. Instead of comparing raw data, we use the metric by exploiting domain-relevant features using an embedding network such as the inception network (Szegedy et al., 2015). The FD of two normal distributions $\mathcal{N}(\boldsymbol{\mu}_1,\boldsymbol{\Sigma}_1)$ and $\mathcal{N}(\boldsymbol{\mu}_2,\boldsymbol{\Sigma}_2)$ is given by:

$$
FD = d^2((\boldsymbol{\mu}_1,\boldsymbol{\Sigma}_1),(\boldsymbol{\mu}_2,\boldsymbol{\Sigma}_2)) = ||\boldsymbol{\mu}_1 - \boldsymbol{\mu}_2||_2^2 + Tr(\boldsymbol{\Sigma}_1 + \boldsymbol{\Sigma}_2 - 2(\boldsymbol{\Sigma}_1\boldsymbol{\Sigma}_2)^{\frac{1}{2}})
$$

While Heusel et al. (2017) show that the distance correlates well with human judgement in the vision domain, it can also be applied to other domains if a domain relevant model replaces the inception network. We apply the FD to measure the mismatch of two sets in the following sections.

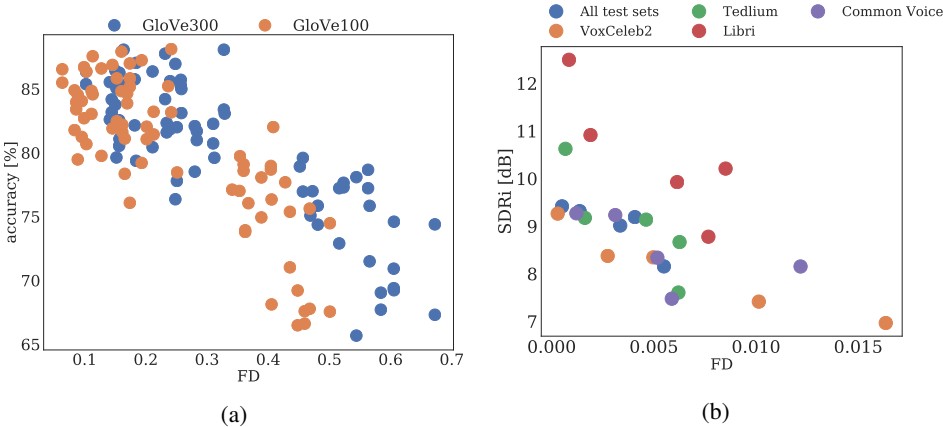

(a)                                          (b)

Figure 3: (a) Sentiment classification accuracy versus FD on the Amazon review dataset. Each point corresponds to a train and test set pair. Two independent runs are displayed with different GloVe embeddings used to compute FD scores. The Pearson correlation is -0.78. (b) Scatter plot of the relation between SDRi and FD scores on the speech separation task. The Pearson correlation between all SDRi and FD points is -0.61, while the average correlation given a fixed test set is -0.82.

## 5 EXPERIMENTS AND RESULTS

Next, we aim to show that the distribution distance between training and test distributions matters in many practical problems. We underline this perspective by running experiments in text classification, vision data augmentation, and speech separation. Crucial for these experiments is that we ensure that the output distribution stays the same while only the input distribution changes. In the following we offer the general experimental setup for each dataset. For more comprehensive implementation details please refer to appendices B and C.

### 5.1 TEXT CLASSIFICATION

We begin our investigation in text classification. To this end we employ different subsets of a large text classification dataset which classifies product reviews into binary good or bad product ratings. To measure the effect of distribution shift, we train our rating classifier on texts of one product category and test it on another. This yields many pairwise train/test distances and performances.

Specifically, we use the Amazon Review dataset (He & McAuley, 2016; McAuley et al., 2015) which contains text reviews and ratings divided into various product categories. In our experiments, we select 9 categories of the 5-core (at least 5 reviews per item) subsets. As input we use the review texts and as output we create binary rating categories of good or bad by splitting the rating categories (1-5) into negative (1-2) and positive (4-5) labels. Reviews with a rating of 3 are removed. To infer the FD score between datasets categories, we use pretrained GloVe (Pennington et al., 2014) embeddings of size 100 and 300. The word embeddings of each sentence are averaged to obtain a single embedding for each review.

From Fig. 3a, we can see that the classification accuracy of the binarized ratings decreases the further the training set is away from the test set as measured by the FD score. The relationship correlates strongly (Pearson correlation -0.78), and notably there is no high accuracy for large FD scores.

### 5.2 VISION CLASSIFICATION

Next we examine the effect of distribution shift in vision classification tasks. Special about this problem is that the task is rarely the same across many datasets. Instead we study distribution shifts that are created when applying data augmentation techniques (Leen, 1995).

We study the problem on the popular CIFAR10 (Krizhevsky et al., 2009), Street View House Numbers (SVHN) (Netzer et al., 2011), and Fashion MNIST dataset (Xiao et al., 2017). As a classifica-

Table 1: Classification accuracy and FD scores on the CIFAR10 (C10), SVHN, and Fashion MNIST (FM) data sets. Compared are various data augmentation techniques and their effect on classification accuracy and distribution distances.

| Training Set | C10 | C10 FD | SVHN | SVHN FD | FM | FM FD |
|---|---|---|---|---|---|---|
| No Augmentations | 81.71% | 3.34 | 93.89% | 13.96 | 93.00% | 1.50 |
| + Flipping | 83.12% | 3.31 | 93.13% | 13.93 | 93.35% | 1.87 |
| + Contrast | 81.36% | 3.92 | 94.21% | 14.27 | 93.22% | 1.50 |
| + Contrast + Flipping | 84.15% | 3.88 | 93.09% | 14.01 | 93.40% | 1.85 |
| + Salt and Pepper Noise | 77.19% | 131.82 | 94.20% | 217.96 | 92.73% | 226.51 |
| Crop | 84.24% | 3.34 | 95.69% | 11.85 | 93.15% | 3.16 |
| + Flipping | 85.77% | 3.35 | 95.18% | 11.23 | 92.88% | 3.44 |
| + Contrast | 85.00% | 4.32 | 95.90% | 11.59 | 92.74% | 3.19 |
| + Contrast + Flipping | 86.36% | 4.31 | 95.25% | 11.69 | 93.13% | 3.47 |
| + Salt and Pepper Noise | 81.55% | 125.62 | 95.46% | 194.43 | 91.53% | 216.93 |
| Crop + Resize | 79.77% | 20.99 | 90.08% | 22.89 | 89.83% | 17.77 |
| + Flipping | 79.74% | 20.91 | 88.94% | 22.63 | 89.72% | 18.19 |
| + Contrast | 79.07% | 22.78 | 90.23% | 23.77 | 89.09% | 17.74 |
| + Contrast + Flipping | 80.34% | 22.68 | 89.32% | 23.59 | 89.70% | 18.29 |
| + Salt and Pepper Noise | 78.08% | 224.61 | 90.35% | 266.05 | 88.79% | 255.05 |

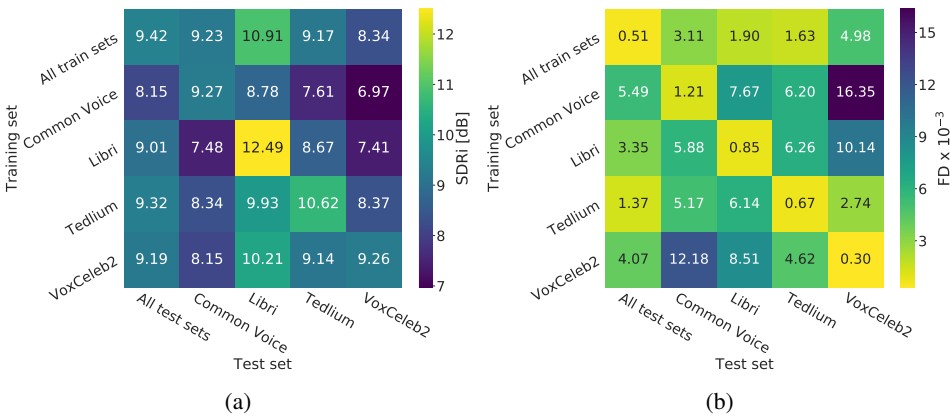

(a)                                    (b)

Figure 4: (a) SDR improvement scores for multiple training and test set configurations. (b) Pairwise FD scores between training and test sets. The matching scatter plot and correlations are reported in Fig. 3b.

tion network, we apply a simple convolutional neural network (LeCun et al., 1998) (see appendix B for more details) and use the Inception-v3 (Szegedy et al., 2015) network to compute the FD score.

In order to create distribution shifts, we apply data augmentation techniques such as flipping, changing contrast, random cropping, and adding salt and pepper noise. We report the resulting classification accuracies and FD scores in Tab. 1.

Intriguing about the results displayed in Tab. 1 is that yet again we see that larger FD scores lead to worse performance. The numbers however are more nuanced than in the previous text experiments. The example of salt and pepper noise shows how nonlinear the relationship between distribution distance and performance can be. While the performance does decrease with increasing distance, it does so at a different scale than observed for the other data augmentation techniques. Nevertheless, these results suggest a direct application of our insights: Selecting the augmentation methods applied based on the FD score between the augmented training set and a validation set. We leave an empirical validation of this use case to future work.

### 5.3 SPEECH SEPARATION

As a final experiment, we venture into speech separation which deals with separating the speech of 2 or more speakers from audio data. This task is unusual because, through the combinatorial way the dataset is generated, it allows for very large datasets. This lets us gauge the influence of distribution shift especially when more than sufficient data is available. Supervised speech separation requires mixed speaker samples of at least two speakers and the corresponding single speaker streams as labels. In the monaural (single-channel) case, one can simply add two single speaker streams to create overlap. The resulting combinatorial possibilities create large datasets.

We collect data from the Libri Speech Corpus (Panayotov et al., 2015), the Voxceleb2 data set (Chung et al., 2018), Common Voice data set (Firefox, 2019) and the TED-LIUM speech corpus (Hernandez et al., 2018). We balance the number of speakers and time per speaker for the first four data sets, keeping 707 speakers for each data set and 7.5 minutes per speaker leading to roughly 88 hours in the training set. For the test set, we sample 10 hours from each data set, from a held back set of data with new speakers. We create mixes from equal datasets since they have more consistent recording settings, which resemble real world situations more closely. We adopt the strongest performing configuration of the Conv-TasNet model (Luo & Mesgarani, 2018) with an SDRi score of 15.6 on the 2-mix WSJ (Hershey et al., 2015) dataset. Identically to Luo & Mesgarani (2018), we use 4 second long samples with a sampling rate of 8k Hz. We report scores as a SDR improvement score (SDRi), obtained through the difference between the SDR (Vincent et al., 2006) of the processed and unprocessed sample. In place of the inception network, we leverage the encoded features of a speaker embedding network to compute FD scores. Detailed hyperparameters and layer configurations are reported in the appendix in Tab. 2.

Again, as depicted in Fig. 4, we find a strong correlation (-0.61) between loss in performance and distribution distance between different training and testing distribution pairs. It appears that larger data sets resulting in better performance is not the whole story. Fig. 4a shows that a large combined dataset performs worse on the individual test sets than a model trained only on the corresponding training set. The presented experiments substantiate theorem 1 by showing a strong correlation between network performance and data distribution distances, especially visible in Fig. 3.

## 6 DISCUSSION AND CONCLUSION

Following popular wisdom, one would be led to believe that more data is all you need. The presented theory and experiments however clearly detail that, while the amount of data is important, ensuring that train and test distribution are close may be similarly significant to perform well on the test set.

From the small-scale and real-world experiments we are left with the startling observation that, frequently, neural networks do not find the "true" functional relationship between input and output. If this were the case, distribution shifts between training and testing should have a smaller impact. Whether this problem can be remedied by finding richer function classes or whether it may be inherently unsolvable will have to be investigated.

An important aspect of this work is how we measure distribution distances. By using a representation network, we obtain low dimensional embeddings and reduce the effect of noisy data. This embedding is however in itself limited by its features, training data, and objective. To apply the insights of this work, it will therefore be paramount to carefully choose an embedding for each dataset and task, whose features are able to meaningfully model the various data shifts a desired learning algorithm would react to. As an example, a word embedding trained only on English texts will not provide meaningful results on other languages and hence is useless for modeling a distribution shift.

Through this work, we emphasize the consequences of using models for predictions, which do not share the invariances of the true functional relationship. In this case, data distribution shifts lead to a deterioration of performance. As a remedy to this issue, we propose applying the Fréchet distance to measure the distance of the dataset distributions to infer the degree of mismatch. With this measure, we can deduce important criteria to choose training sets, select data augmentation techniques, and help optimize networks and their invariances. We believe that making the problem explicit and having a way to measure progress through the FD score may allow for a new wave of innovative ideas on how to address generalization under data shifts.

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

APPENDIX

The appendix comprises extended details of the work that may add to the understanding without being critical to follow the main text.

## A    SMALL-SCALE EXPERIMENTS

The following specifies the experimental details of the small-scale insight experiments from section 3. Throughout these experiments, a model needs to approximate a function $f(x)$ given a point $x \in \mathbb{R}$, which is sampled from the uniform distribution. We limit sampling of $x$ during training to a specific interval $[a, b]$ and add gaussian noise from $\mathcal{N}(0, 0.4)$ to the training labels. We further elaborate the experiments based on the rows of Fig. 1.

In rows one and two of Fig. 1, a model is trained to approximate the function $f(x) = e^{-x}\cos(2\pi x)$ on the training interval $[-2, 2]$. During testing, we shift the test interval by the scalar $\alpha = [0, -0.5, 0.5]$ (images from left to right respectively). Additionally, we apply a zero mean normalization to the input data in the experiments of the second row. Note that the ground truth is invariant to a change in the input distribution and thus stays consistent.

In the third row of Fig. 1, $x$ is drawn from Uniform$[-6, 6]$ during training, while the test set samples lie outside of the training domain in $[-10, -6]$ and $[6, 10]$. The functions from left to right are given as follows:

$$f(x) = 0.006x^3 - 0.027x^2 - 0.018x + 0.015$$

$$f(x) = \begin{cases} -x - 4 & x \leq -2 \\ x & -2 < x < 0 \\ -2x & 0 \leq x < 3 \\ 1.5x - 10.5 & x \geq 3 \end{cases}$$

$$f(x) = -0.2x + 1$$

representing a nonlinear, piecewise linear, and linear function.

The last experiments approximate $f(x) = e^{-x}\cos(2\pi x)$ with changing training domains while keeping the test domain consistent. The training intervals from left to right are $[0, 1]$, $[-1, 1]$, and $[-2, 2]$ with the test interval set to $[-1, 1]$.

For all task we use the same neural network as a model. We train a two-layer feedforward neural network with 8 hidden units in its first layer followed by a ReLU (Nair & Hinton, 2010) activation function and a linear layer with one output unit in its second layer. In total the network consists of 25 parameters including biases, which we optimize using the mean squared error and Adam optimizer (Kingma & Ba, 2014).

## B    TEXT CLASSIFICATION

To complete the description of the experiment in section 5.1, we offer a more detailed description of the preprocessing and network details. Before a model can use the text data from the amazon review dataset, several processing steps are necessary to convert text to numeric representations. Each review contains several information fields. However, we only utilize the overall score and the review text, from which we remove capitalization, punctuation and multiple digit numbers. The overall score (1-5) is split into positive (4-5) and negative (1-2) reviews, while ratings of 3 are removed. The vocabulary size is limited to 10,000 words resulting in less frequent words being replaced by the $< \text{UNK} >$ symbol. To achieve consistent sentence lengths, we cut or pad reviews to a fixed length of 256 with padded words set to the symbol $< \text{PAD} >$. As a last processing step, we tokenize the training data and apply the mapping to the paired test set. We report the complete classification accuracies and FD scores in Tab. 3 between each category used as a training and test set.

The classification network consists of an embedding, a global average pooling, and three dense layers of size 32, 16 and 1, respectively. ReLUs (Nair & Hinton, 2010) are applied after the first two

dense layers and a Sigmoid activation function is applied to the last layer. We use Adam (Kingma & Ba, 2014) and the binary cross-entropy loss for optimization with a learning rate of 0.01.

## C  VISION CLASSIFICATION

Again we offer complementary details to the vision classification experiments with data augmentation. Especially, we provide more details on the examined datasets, augmentations, and classification network. The CIFAR10 dataset (Krizhevsky et al., 2009) consists of $32 \times 32$ colored natural images divided into 10 classes. The training and test set consists of 50,000 images and 10,000 images, respectively. The Street View House Numbers (SVHN) (Netzer et al., 2011) dataset, also contains 10 categories, one for each digit between 1 and 10. For SVHN, training set and test set consist of 73,257 images and 26,032 images, respectively. The Fashion MNIST dataset (Xiao et al., 2017) consists of 60,000 train and 10,000 test images. The dataset contains grayscale images of clothes again classified into 10 classes.

For data augmentation we offer complementary details. Random cropping reduces the image size to $24 \times 24$ and is only applied to the middle of images in the test set. We differentiate between only cropping and cropping and resize, in which the images are rescaled back to $32 \times 32$. However, the test set remains unchanged when rescaling. For salt and pepper noise, we set distorted pixels equiprobably to black or white with a probability of 33%. For random contrast adjustment, we set the upper and lower limit to 1.8 and 0.2, respectively.

The classification network is based on a CNN (LeCun et al., 1998) structure. Convolutional layers of size 64 use a kernel size of $5 \times 5$ followed by a ReLu (Nair & Hinton, 2010) activation function, a max pooling layer with a kernel size of $3 \times 3$ and stride 2, and a local response normalization (Krizhevsky et al., 2017a) with a depth radius of 4, a bias of 1, $\alpha$ equal to $\frac{0.001}{9}$, and $\beta$ set to 0.75. After two blocks three fully-connected layers of size 384, 192 and 10 are used, with ReLu activations (Nair & Hinton, 2010) after the first two and a softmax layer following the last layer. For optimization, we use the cross-entropy loss and gradient descent (Robbins & Monro, 1951) with a initial learning rate of 0.1 and an exponential decay rate of 0.1. Additionally, we use exponential moving average on the weights with a decay factor of 0.9999.

In order to compute FD scores, we use the Inception-v3 (Szegedy et al., 2015) network, which is available pretrained in the Tensorflow library (Abadi et al., 2016). Images are resized to $299 \times 299$ to fit the input dimension of the network, which was pretrained on Imagenet (Deng et al., 2009). The resulting feature representations are retrieved from the pool-3 layer with 2048 output dimensions.

## D  SPEECH SEPARATION

Making the description of the speech separation experiments more comprehensive, we offer the following additional implementation details for the speech embedding network for which we adopt the generalized end-to-end loss (GE2E) proposed by Wan et al. (2017). We train a network build on convolutional and LSTM (Hochreiter & Schmidhuber, 1997) layers. The inputs are based on 3 second long samples transformed into the Short-Time Fourier Transform (STFT) spectrogram and the Mel-frequency cepstral coefficients (MFCC). For the spectrogram creation, we use a hop size of 6 ms and a window length of 23 ms. We apply a simple 1-dimensional convolutional layer with a kernel size of 1 to the MFCC to match the number of frequency bins of the STFT, such that we can use the concatenation of both as an input to our network. Tab. 2 summarizes the network following the input, predicting a 100 dimensional speaker embedding.

Table 2: Detailed specifications of the speaker embedding network trained with the GE2E Loss (Wan et al., 2017) applied to calculate FD scores in section 5.3. The network is based on the densely connected network (Huang et al., 2016), fully-connected layers (FC), LSTMs (Hochreiter & Schmidhuber, 1997), and Statistical Pooling (Snyder et al., 2017). As an optimizer, we use Adam (Kingma & Ba, 2014) with a learning rate of $0.0001$.

| Layer | Filter Sizes | Filters | Additional Specifications |
|---|---|---|---|
| Dense Block | $\begin{bmatrix} 1 \times 1 \\ 3 \times 3 \end{bmatrix}$ | $\begin{bmatrix} 64 \\ 16 \end{bmatrix} \times 2$ | densely connected |
| CNN | $3 \times 3$ | 32 | stride of 2 |
| Dense Block | $\begin{bmatrix} 1 \times 1 \\ 3 \times 3 \end{bmatrix}$ | $\begin{bmatrix} 64 \\ 16 \end{bmatrix} \times 2$ | − |
| CNN | $3 \times 3$ | 64 | stride of 2 |
| Dense Block | $\begin{bmatrix} 1 \times 1 \\ 3 \times 3 \end{bmatrix}$ | $\begin{bmatrix} 64 \\ 16 \end{bmatrix} \times 12$ | − |
| CNN | $3 \times 3$ | 128 | stride of 2 |
| Dense Block | $\begin{bmatrix} 1 \times 1 \\ 3 \times 3 \end{bmatrix}$ | $\begin{bmatrix} 64 \\ 16 \end{bmatrix} \times 12$ | − |
| CNN | $3 \times 3$ | 256 | stride of 2 |
| Dense Block | $\begin{bmatrix} 1 \times 1 \\ 3 \times 3 \end{bmatrix}$ | $\begin{bmatrix} 64 \\ 16 \end{bmatrix} \times 12$ | − |
| CNN | $3 \times 3$ | 512 | stride of 2 |
| Dense Block | $\begin{bmatrix} 1 \times 1 \\ 3 \times 3 \end{bmatrix}$ | $\begin{bmatrix} 64 \\ 16 \end{bmatrix} \times 12$ | − |
| CNN | $3 \times 3$ | 1024 | stride of 2 |
| $\begin{bmatrix} \text{CNN} \\ \text{LSTM} \end{bmatrix}$ | $\begin{bmatrix} 1 \times 1 \\ - \end{bmatrix}$ | $\begin{bmatrix} 512 \\ 512 \end{bmatrix} \times 5$ | − |
| Stats Pooling | − | − | with max, mean, variance features |
| + CNN | 3 | 200 | − |
| + FC | − | 100 | with flattened time dimension |
| Stats Pooling | − | − | applied on the last strided CNN output |
| + CNN | 3 | 200 | − |
| + FC | − | 100 | flattening the input beforehand |
| Addition | − | − | on both stats pooling results after the FC layer |
| L2 normalization | − | − | − |

Table 3: Detailed accuracy and FD on the Amazon review categories. Models are trained by using different categories as train and test pairs. The FD is obtained with the GloVe100 embedding.

| Training Set / Test Set | Electronics | Home and Kitchen | Digital Music | Sports and Outdoors | Toys and Games | Movies and TV | Clothing Shoes Jewelry | Office Products | Pet Supplies |
|---|---|---|---|---|---|---|---|---|---|
| **Training Set** | | | | | Accuracy | | | | |
| Electronics | – | 87.92% | 76.34% | 86.71% | 87.0% | 79.73% | 88.12% | 84.54% | 82.04% |
| Home and Kitchen | 84.83% | – | 75.62% | 85.49% | 86.88% | 78.69% | 85.83% | 84.84% | 81.78% |
| Digital Music | 68.12% | 67.78% | – | 71.02% | 73.79% | 79.2% | 67.56% | 66.61% | 66.49% |
| Sports and Outdoors | 82.7% | 86.54% | 75.37% | – | 86.6% | 79.09% | 87.56% | 84.05% | 83.4% |
| Toys and Games | 76.08% | 81.91% | 73.93% | 79.75% | – | 78.46% | 83.87% | 82.2% | 80.68% |
| Movies and TV | 77.11% | 78.97% | 87.24% | 78.58% | 85.22% | – | 82.01% | 78.06% | 77.01% |
| Clothing Shoes Jewelry | 83.16% | 85.18% | 74.48% | 84.58% | 84.65% | 77.69% | – | 83.21% | 82.47% |
| Office Products | 79.47% | 83.04% | 67.59% | 81.24% | 81.62% | 74.94% | 81.43% | – | 78.35% |
| Pet Supplies | 81.06% | 84.89% | 69.21% | 83.96% | 86.36% | 76.05% | 85.83% | 81.11% | – |
| **Training Set** | | | | | FD | | | | |
| Electronics | – | 0.1602 | 0.4055 | 0.099 | 0.1739 | 0.3539 | 0.2415 | 0.0888 | 0.2014 |
| Home and Kitchen | 0.1602 | – | 0.4678 | 0.0635 | 0.1455 | 0.4044 | 0.174 | 0.1113 | 0.0836 |
| Digital Music | 0.4055 | 0.4678 | – | 0.4355 | 0.3624 | 0.1932 | 0.5008 | 0.4596 | 0.4481 |
| Sports and Outdoors | 0.099 | 0.0635 | 0.4355 | – | 0.1275 | 0.3598 | 0.1132 | 0.0954 | 0.0862 |
| Toys and Games | 0.1739 | 0.1455 | 0.3624 | 0.1275 | – | 0.2511 | 0.1695 | 0.1616 | 0.1029 |
| Movies and TV | 0.3411 | 0.4044 | 0.1932 | 0.3598 | 0.2366 | – | 0.4082 | 0.3888 | 0.353 |
| Clothing | 0.2415 | 0.174 | 0.5008 | 0.1132 | 0.1695 | 0.4278 | – | 0.2128 | 0.1526 |
| Office Products | 0.0888 | 0.1113 | 0.4596 | 0.0954 | 0.1616 | 0.3888 | 0.2128 | – | 0.1656 |
| Pet Supplies | 0.2014 | 0.0836 | 0.4481 | 0.0862 | 0.1029 | 0.3676 | 0.1526 | 0.1656 | – |

