# OpenReview forum: "The Frechet Distance of training and test distribution predicts the generalization gap"
_ICLR.cc/2020/Conference — Reject_

### Official Review · AnonReviewer3 · 2019-10-23
**Official Blind Review #3**

**Rating:** 3

**Review:**

The authors consider the relation between Frechet distance of training and test distribution and the generalization gap. The authors derive the lower bound for the difference of loss function w.r.t. training and test set by the Wasserstein distance between embedding training and test set distribution. Empirically, the authors illustrate a strong correlation between test performance and the distance in distributions between training and test set.

The motivation to find the relation between generalization gap and the Frechet distance of training and test distribution is sound. However, I am not sure that the lower bound as in Equation (1) is enough. I am curious that one can derive the upper bound for the relation or not. The finding about choosing a training data distribution should be close to the test data distribution seems quite trivial in some sense. I am not clear about its important since it is quite popular that the distribution shift affects the performance and many learning approach assumes same distribution for training and test data. Overall I feel that the contribution may be quite weak, and I lean on the negative side.

Below are some of my concerns:

1) About the lower-bound in Equation (1), it seems unclear to me that when the W_2(p1, p2) = 0, we can inference any information about the test performance (It seems quite trivial for this case, the left hand side time is greater than or equal 0?) In my opinion, the upper-bound is more important which one can inference much information about the difference of generalization gap.

2) In the proof of Theorem 1, it is quite hard to follow with the current notation, for the integral in (i), (ii) as well as in the proof using the intermediate value theorem, which variables are used? I am confused which one is variable, which one is constants in those integrals.

3) In page 5, at the interpretation (1), for W2(p1, p2) = 0, the learned function fits training distribution perfectly and is not ill-conditioned ==> why one can deduce that the test distribution is fit perfectly? What we have in Theorem 1 is the lower-bound only?



**Experience Assessment:**

I have read many papers in this area.

**Review Assessment: Checking Correctness Of Derivations And Theory:**

I assessed the sensibility of the derivations and theory.

**Review Assessment: Checking Correctness Of Experiments:**

I assessed the sensibility of the experiments.

**Review Assessment: Thoroughness In Paper Reading:**

I read the paper at least twice and used my best judgement in assessing the paper.

---

### Official Review · AnonReviewer2 · 2019-10-23
**Official Blind Review #2**

**Rating:** 3

**Review:**

The authors propose to relate the performance of a classifier under distribution shift using a quantity called Frechet distance. It is common belief that the further apart the training and test distributions are, the more difficult it is to transfer a learned classifier. They give simple bounds via gradient norm/Lipschitz constants and distribution distance in Theorem 1. The authors try to capture it with Frechet distance, but I struggle to understand what is new in this work.

First, there are a lot of assumptions in the computation of the Frechet distance:
  1. The authors use the embeddings given by the neural networks instead of the raw data since density estimation is hard. This makes the distance model-dependent
  2. The authors assume the embeddings are normally distributed in their computation, which have not been justified.

Most importantly, they do not relate the Frechet distance to the lower bound in Theorem 1. There is no estimation on how the learned changes across distributions in the gradient norm term. This makes the evaluation nothing more than a confirmation of the general idea that the closer the distribution, the better the transfer. The lower bound is not used in any quantitative manner.

The authors should make the connection of the bound and its computation clear, with proper connections to the experiments. The current paper looks like separate theoretical and experimental results that do not tie together.



**Experience Assessment:**

I have published one or two papers in this area.

**Review Assessment: Checking Correctness Of Derivations And Theory:**

I assessed the sensibility of the derivations and theory.

**Review Assessment: Checking Correctness Of Experiments:**

I assessed the sensibility of the experiments.

**Review Assessment: Thoroughness In Paper Reading:**

I read the paper thoroughly.

---

### Official Review · AnonReviewer1 · 2019-10-24
**Official Blind Review #1**

**Rating:** 3

**Review:**

This paper considers the problem of how the mismatch between distributions of training data and test data would affect the generalization gap in machine learning tasks. This phenomenon has been observed many times in previous literature and has gathered significant attention in the machine learning community.

The paper took a step in relating the change in the performance of the learned function to the Frechet distance (FD), also known as 2-Wasserstein distance, between the input and output distributions and proved that the former is lower bounded by the latter multiplied by a term related to the sensitivity of learning algorithm to distribution shift. The paper also provides empirical evidence that the testing error is correlated with the FD between input and output distributions based on tasks including text classification, image classification, and speech separation.

I find the idea of the paper interesting but the content not convincing enough. The theory proved in the paper does not provide additional quantitive insight beyond intuition. Specifically, the term about the sensitivity of the algorithm is not justified enough in the paper. The experiments provide some evidence but not convincing, especially for the part about image classification.

I also find the statement about the generalization gap a bit misleading. Generally, the generalization gap refers to the gap between the expected error and the empirical error.  But the experiments are mostly presenting the performance on the test data.

Overall, I don't think the paper meets the standard for publication at ICLR.

**Experience Assessment:**

I have read many papers in this area.

**Review Assessment: Checking Correctness Of Derivations And Theory:**

I assessed the sensibility of the derivations and theory.

**Review Assessment: Checking Correctness Of Experiments:**

I assessed the sensibility of the experiments.

**Review Assessment: Thoroughness In Paper Reading:**

I read the paper at least twice and used my best judgement in assessing the paper.

---

### Decision · Program_Chairs · 2019-12-19

**Decision:**

Reject

**Comment:**

The authors discuss how to predict generalization gaps. Reviews are mixed, putting the submission in the lower half of this year's submissions. I also would have liked to see a comparison with other divergence metrics, for example, L1, MMD, H-distance, discrepancy distance, and learned representations (e.g., BERT, Laser, etc., for language). Without this, the empirical evaluation of FD is a bit weak. Also, the obvious next step would be trying to minimize FD in the context of domain adaptation, and the question is if this shouldn't already be part of your paper? Suggestions: The Amazon reviews are time-stamped, enabling you to run experiments with drift over time. See [0] for an example.

[0] https://www.aclweb.org/anthology/W18-6210/